# Understanding the Impact of Waste Management on a Destination's Image: A Stakeholders' Perspective

**Aglaia-Spyridoula Koliotasi [1], Konstadinos Abeliotis [2,*] and Paris-Georgios Tsartas [2]**

1 MSc "Sustainable Tourism Development, Heritage, Environment, Society", Harokopio University of Athens, 176 76 Kallithea, Greece
2 Department of Economics and Sustainable Development, Harokopio University of Athens, 176 76 Kallithea, Greece
* Correspondence: kabeli@hua.gr; Tel.: +30-21-0954-9363

**Abstract:** The present manuscript describes a case study on the viewpoints of tourism stakeholders on the effect of waste management on a destination's image. In particular, the study aims to analyze the problems that arose during the summer of 2018 in the touristic image of Corfu because of the waste management on the island. The qualitative method approach through interviews was used to collect the primary data of the survey; online sources were used to collect data to review similar cases. All the stakeholders agree that waste management during the summer of 2018 negatively affected the image of Corfu as a touristic destination. Moreover, based on the interviews' findings, there is a lack of political will, infrastructure, and information provision from the local authorities on the island of Corfu. Interviews also indicated that an integrated solution to the problem of waste management on the island is the implementation of source separation in households and tourism-related firms with the support of the local authorities. The contribution of this paper is towards identifying the effect that waste management has on the image of a tourist destination. It is the first of its kind conducted in Greece and among the few reported in the literature focusing on the viewpoints of service providers.

**Keywords:** sustainability; island; Corfu; Greece

## 1. Introduction

Tourism can negatively affect the local environment of a destination because of the visitors' activities [1]. Solid waste management is among the factors that can negatively affect the local environment of a tourist destination. Therefore, inefficient solid waste management may reduce the tourist value of an otherwise attractive location [2]. The challenge of proper waste management becomes crucial in touristically developed islands since waste management systems have to deliver efficient services in a short period due to the area limits set by the insular character. Many islands face a crucial problem regarding waste management [3]. According to Santamarta et al. [4], many islands in the world (e.g., Hawaii, Malta, Cyprus, Mallorca, Azores, the islands of the Caribbean, etc.) face problems regarding waste. Islands are usually densely populated and most of them are considered tourism destinations [3], and this results in the large generation of waste [5]. As Karkazi et al. [6] stated, it is difficult for islands to manage waste properly and in a sustainable way because of the increase in population during the summer months and the lack of available land.

The manuscript aims to present the viewpoints of various tourism service providers in Corfu regarding the effect that poor waste management had on the touristic image of the island of Corfu during the summer of 2018. The manuscript is of interest for tourist researchers and practitioners since it discusses the effect of a current global issue (i.e., solid waste management) on the image of touristic destinations.

The manuscript consists of the following sections: a literature review on waste management issues of islands is presented first; then, a brief outline of the situation of waste

management on the island of Corfu is presented; and the methodology, results, and conclusions of the research are then presented.

## 2. Literature Review

According to the United Nations World Tourism Organization [7], islands are the most visited destinations by tourists every year. There are many advantages as well as disadvantages to the development of an island as a tourism destination. It has been identified by various authors that tourism aggravates the waste management problem [8–12]. Due to their isolation and the intensive tourism industry, islands seem to present the highest per capita waste generation indicators [13]. Moreover, tourism activities, since they follow a seasonal pattern, increase the seasonal quantities of waste generated; they also alter the seasonal qualitative composition of the waste stream on islands [10].

Eckelman et al. [14] identified six common barriers to waste management on islands. The authors conclude that island societies face a diverse set of waste management challenges that have serious consequences for the health of their ecosystems, economies, and citizens.

Martins and Cró [15] argue that environmental resources are inputs of production in the creation of the tourist experience. Therefore, there may be negative impacts of improper solid waste management on the image of a destination [15]. They also state that the tourism sector is particularly intensive in solid waste generation and that the incoming tourists constitute an additional source of solid waste generation in the tourist destination. The authors propose that tourists should be charged between EUR 0.88 and EUR 0.98 per overnight stay in Madeira, Portugal to negate the extra cost of solid waste disposal [15]. Charging the tourists a unit-based fee for waste management by local governments is also proposed by Manomaivibool [16].

Moreover, insular municipalities often face difficulties managing solid waste because of the lack of infrastructure, financial resources, complexity, and system multidimensionality [17]. The problem that an island faces due to improper waste management is more critical because of the limited area of the island [5,18]. Furthermore, it is difficult for small islands to find markets to sell recyclables on the mainland [5,8,19]. In addition, in contrast with the continent, islands have a reduced number of treatment and disposal facilities and significant seasonality due to tourism [17]. Small islands manage the waste by dumping or burning it, while some islands promote recycling [5,8,18].

A recent case study from the Canary Islands in Spain argues that sustainable waste management is not yet as advanced as other environmental practices [20]. The authors also recommend that hospitality stakeholders should reflect upon their knowledge and experience to better comprehend the contribution of tourism services to waste generation [20].

Improper waste management affects the sustainability of the destination's environment, the quality of the tourism experiences that the destination offers, and the quality of life of the host community [20,21]. If the local municipalities could manage the waste properly, excessive waste generation would not be a problem [5,9].

The EU's tourism sector generates 35 million tons of solid waste per year, representing 7% of the total waste generation of the services sector. As Shamshiry et al. [19] stated, tourism waste generation is almost twice the rate of local waste generation. The fact that, in certain destinations, the waste generation rate per guest is reported to be twice as much as the waste generation rate per local capita is also reported by Manomaivibool [16]. Focusing on Greece, the Greek hotel industry, together with the cruise sector, is estimated to produce 400,000–550,000 tons of solid waste with the average (per night per person) ranging from 1.7 to 2.5 kg. Meanwhile, food and beverage services generate up to 650,000 tons a year [22]. The composition of solid waste from the hospitality sector in Greece is mainly organic (e.g., food waste) and has a large share of packaging materials [10]. Aiming to motivate all Greek tourism enterprises to implement environmentally friendly practices, the Greek Tourism Confederation's research department INSETE issued a handbook in 2018 on "Recycling and Solid Waste Management" [22].

### 3. A brief Timeline of the Waste Management on the Island of Corfu

Corfu, an island of 610.9 Km$^2$, located in the Ionian Sea in the northwest part of Greece, is one of the main touristic hotpots of Greece. The difficulty in managing solid waste in Corfu was perceived from the 1960s when tourism increased rapidly and so did the volume of solid waste generated. As a result, uncontrollable dumps were created, and this non-sustainable management practice escalated the problem. Skordilis [23] reports the operation of 19 landfills in Corfu during the 1990s, 18 of which did not operate following the legal sanitary specifications. Then, the establishment of two sanitary landfills was proposed, one in the south part of the island, and one a few kilometers away from the city of Corfu in a small village called Tebloni.

The sanitary landfill of Tebloni is situated in central Corfu, and it started operating in 2003. The landfill never operated in agreement with the requirements of the European Union because the generated biogas was not collected, and the leachates were not treated. Overall, the landfill did not fulfill the planning requirements, to such an extent that it looked more like a dump than a sanitary landfill. For all the shortcomings mentioned above, in 2015, the European Commission referred Greece to the Court of Justice of the EU over inefficient waste management on the Greek island of Corfu [24]. The Commission was concerned that the Tebloni landfill, operating in breach of EU waste and landfill legislation since at least 2007, represented a severe risk to human health and the environment [24]. The operating failure of the Tebloni landfill shifted the managerial focus to the landfill situated in Lefkimmi, in southern Corfu.

The sanitary landfill in Lefkimmi was constructed in 2007 and was planned to operate for 20 years with an average input of 10,000 tons of waste per year. However, the Lefkimmi landfill did not ever operate because of the conflict between the residents and the Municipality of Corfu, a typical expression of the not-in-my-backyard syndrome. In July 2018, during the peak tourist season, the Corfu waste management system collapsed since there were no available operating landfill sites on the island. A waste crisis was declared since all the waste volumes were left uncollected and/or on the streets of the island (see Figure 1). Ugly waste images were reported for Corfu in major networks [25,26]. The European Commission "was calling on Greece to ensure that EU waste legislation is implemented on the ground in Corfu Island" [27]. The local tourist stakeholders were alert but also left with no hope.

The political decision to alleviate the impacts of the failure of solid waste management on the island of Corfu was to transport the waste to the sanitary landfill of Kozani, located 250 km away from Corfu, in the northern part of mainland Greece. Shipping waste from islands to the mainland has also been reported as very costly, but the only feasible option once other alternatives are exhausted [5].

This manuscript aims to explore the effect of ineffective waste management on the island of Corfu during 2018, and on its image as a tourist destination based on the experiences of local tourist stakeholders. This was the starting point for undertaking a field research study, the results of which are presented in the following sections. Crompton [28] defined destination image as the sum of tourists' beliefs, ideas, and impressions of a destination. The perceptions about a destination are referred to as destination image [29]. Hankinson [30] reported on the relative saliency of destination image attributes associated with the history, heritage, and culture in shaping the perceptions of places as tourism destinations. Such organic images tend to have been formed over a long period [30]. However, other images, such as ugly images of uncollected waste in an insular setting, may have a negative result on a destination's image, due to communication processes that fall within marketing's goals. Such favorable images of a destination are among the factors that can lead to return visits and recommendations [31] and the perception of a place as a sustainable destination [32].

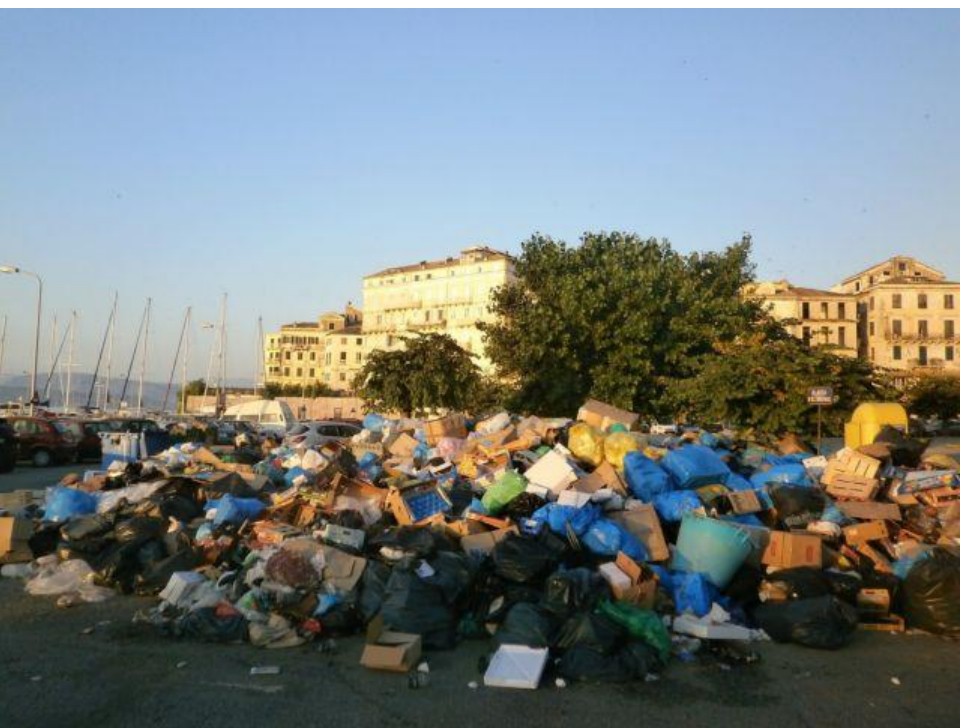

**Figure 1.** The waste situation in Corfu during the summer of 2018.

## 4. Materials and Methods

For the primary data collection of this case study, a qualitative research method was chosen. A case study is a "methodology based on interviews, which are used to investigate technical aspects of a contemporary phenomenon within its real-life context" [33]. As Bryman [34] underlined, the interview is the most frequent and well-known method of data collection in qualitative research. According to Stark and Torrance (2005) [35], interviews are suitable when choosing a case study.

The interviewees were various stakeholders involved in the tourism sector on the island of Corfu. The researchers separated stakeholders into three groups: (i) hotel and restaurant owners, (ii) tour operators, and (iii) political decisionmakers and experts on waste management. The complete list of the stakeholders is presented in Table A1 (see Appendix A).

Following an initial telephone contact, the researcher sent the questions to the participants via email. The interviews, conducted in Greek, were recorded and transcribed by the main researcher. All the interviewees signed a consent form on the use of their responses in the research. A total of 16 interviews were conducted, and each interview lasted approximately 30–40 min. The discussions took place between July and August 2020 on the island of Corfu. For the needs of the current manuscript, all the responses were translated into English.

The interviewees were asked to express their viewpoints on specific open-ended questions for additional questions to be raised [34]. The questions were divided into three main thematic sections: the first one was about the general waste management situation in Corfu and the respective perspectives of the stakeholders. Stakeholders were also asked about their opinion on the municipal waste management services. Then, there were questions on the issue of source separation of waste. Finally, there were questions about the stakeholders' attitudes regarding the effect of inefficient waste management on Corfu's destination image.

## 5. Results

### 5.1. The Effects of Inefficient Waste Management on Corfu′s Destination Image

A hotel manager (stakeholder 9) stated that it was complicated for him to explain to the visitors that they were safe and that the hotel was clean, despite the uncollected waste in the surroundings. Another hotel owner (stakeholder 7) believes that Corfu′s destination image has been damaged even though the number of visitors is still high. However, he stated that the quality of the visitors is not the same. It is difficult to persuade people to come to a luxury hotel and be confronted with this situation on the island. The chairman of the hotel association of Corfu (stakeholder 6) also stated that Corfu′s destination image had been damaged because of the defamation caused by social media and the internet in general. During his interview, he mentioned that they consulted communications experts, and they responded that a destination needs ten good pictures to negate a single lousy image. So, as a hotel association, they ran a campaign in cooperation with "Marketing Greece". A hotel manager (stakeholder 10) also supported the argument that although Corfu has a firm destination image, visitors were disappointed with the situation that they faced. He also stated that if the problem persists, Corfu′s touristic activity will be affected.

Furthermore, a restaurant owner (stakeholder 12) said, "I wonder why they (i.e., tourists) still come". Another restaurant owner (stakeholder 11) added that Corfu′s destination image had been damaged not only because of inefficient waste management but because of mass tourism as well. According to his wording, "Corfu used to be called the pearl of the Mediterranean. Mass tourism ruined this image".

Stakeholder 6, the chairman of the hotel association of Corfu, claimed that if the situation does not improve, Corfu will face problems in the future regarding tourism. However, in his opinion, people forget, so if we provide them with good images, he is sure that Corfu will become a top destination again. Additionally, the tour operator (stakeholder 16) said that the problem regarding waste management had not affected tourism as much as we think yet. He clearly stated that, "foreign tour operators did not care about the situation in Corfu, and there were no cancellations because of the situation with inefficient waste management".

### 5.2. The Attitudes of Stakeholders on Source Separation and Recycling

Stakeholder 2, an expert on waste management, stated, "for a sustainable and permanent solution, source separation is necessary". Stakeholder 8, a former hotel manager, pointed out that source separation will decrease the volumes of mixed waste, while the chairman of the hotel owners of Corfu (stakeholder 6) pointedly said that, "source separation is the ultimate solution to the problem". Stakeholder 1, the chairwoman of the cultural association of Tebloni, pointed out that an effective waste management system should strategically implement a door-to-door collection system and the pay-as-you-throw principle, and offer incentives and penalties to the citizens. Stakeholder 3, a member of the Waste Watch Corfu NGO, agreed with stakeholder 1, and both stated that political will is essential to develop a strategic plan. On the other hand, although stakeholder 4, a local political activist, pointed out the importance of source separation, he emphasized the importance of management of the organic fraction, i.e., waste generated mainly by food leftovers.

When stakeholder 11, a restaurant owner, was asked if he is satisfied with the steps that have been taken regarding source separation, he pointedly and sarcastically answered, "where?". Only specific large-scale touristic organizations voluntarily participate in source separation because they represent foreign investors with increased environmental awareness. Stakeholders 1 and 4 (the representative of the citizens of Tebloni and the local political activist) reported that there are many efforts from citizens and firms. Still, they do not have the support of local authorities. Stakeholder 5, the vice mayor of North Corfu′s municipality, claimed that each municipality is independent to decide. The south and central Corfu municipalities follow the system with the blue bins (i.e., all dry recyclable materials are placed in blue bins). According to stakeholder 2, the waste management

expert, steps have to be taken to increase effective source separation. The first step is to organize actions for materials that are easier to manage, but at the same time, solutions for all recyclable materials should be found.

Stakeholder 9, a hotel manager, claimed that the municipal tax is not reasonable because they also pay EUR 3500–4000 per month to a private company to manage their waste on top of the municipal tax. The municipal tax would be reasonable if there were proper collection and service in general. Stakeholder 11, a restaurant owner, cannot realize why he pays the municipal tax when there are shortages in the provision of water, there is no lighting on the streets, and there is no waste management.

The NGO representative, stakeholder 3, proposed that citizens who contribute to the source separation process should benefit from the value of recyclable materials. However, the waste management expert, stakeholder 2, stated that, "the main motive is to manage our waste properly not to face the same problems we faced in 2017–2019". This notion was supported by some representatives of the hotels who told the researcher that, "hotels are not interested in being benefited from the economic revenue of the recyclable materials; hotels care for their reputation, as organizations that manage the recyclable materials and have an increased environmental awareness". According to stakeholder 11, a restaurant owner, "the role of the local government is to promote recycling, educate people and supervise this attempt because tourism entities cannot organize their system of source separation". Note, however, that the hotel association of Corfu tried to organize an autonomous recycling system. Still, it was challenging due to the topography and the size of the island, as the chairman of the hotel association of Corfu, stakeholder 6, explained.

According to stakeholder 1, source separation is the solution to the problem. As she stated, the municipality of North Corfu is the only municipality that has implemented actions regarding source separation. If the municipality of Central Corfu follows the example of North Corfu and establishes a system with the appropriate infrastructure, the results will be spectacular.

To date, we have identified that the stakeholders associated with the tourism sector are not satisfied with the waste management services that they receive from the local municipalities in Corfu. In the following, we present the perspective of the local political stakeholders. The municipality of Central Corfu hired a consultant, namely stakeholder 2, and according to NGO stakeholder 3, public events have been organized for informing the citizens. More specifically, stakeholder 2, the waste management expert, organized a "waste management" class which was offered free of charge to all citizens. So, all the citizens of Corfu, including the tourism stakeholders, had the opportunity to be informed about the proper way of managing waste. However, stakeholder 2 claimed that tourism professionals are already aware of source separation and recycling benefits. On the other hand, stakeholder 3, a member of the Waste Watch Corfu NGO, mentioned that the municipality does not provide information on the benefits of recycling or the proper way of recycling because it cannot support recycling.

On the question about source material separation, the Vice Mayor of Circular Economy and Environment of the municipality of North Corfu mentioned that there were six "green corners" when she took up the office. Since then, 22 new "green corners" have been established, in which four different materials are being collected separately. According to the Vice Mayor, the quantities of recyclable materials are enormous, and the separation purity of these materials reaches 90–95%. Furthermore, the municipality of North Corfu has organized the collection of recyclable material from the schools and the collection of paper from firms that produce vast amounts of it, such as hotels. Moreover, according to the Vice Mayor, the collection of paper reaches ten tonnes per week. Last, but not least, more than ten thousand information leaflets have been distributed to the residents, in which directions about proper source separation are given. Regarding the next steps, "the municipality will strengthen the infrastructure, hire more personnel, and put in place the brown bin for the separate collection of organic waste".

*5.3. The Responses of the Stakeholders on the Sanitary Landfill of Tebloni and the Construction of a New Solid Waste Treatment Plant*

Stakeholder 1, the chairwoman of the cultural association of Tebloni since 2007, said that in the town of Tebloni, there had been uncontrolled waste dumps since 1987. In 2002, the decision to construct a sanitary landfill was made. The residents did not have any objection because they considered this project as the one that would solve the problem of uncontrolled waste dumps. However, the project was not well organized, and problems emerged immediately. For instance, the combustion of collected biogas was not implemented and the landfill leachate ended up in a nearby lake, one of the seven lakes in the vicinity of the sanitary landfill.

Stakeholder 1, the representative of the residents of Tebloni, was also asked if she agrees with the plan for the restoration of the sanitary landfill in Tebloni, which anticipated the storage of 40,000 tonnes of waste. She stated that such projects require studies about the suitability of the place and that the government spent EUR 600,000 for such a study but local stakeholders do not have access to the results of this study. She was also asked if the residents of Tebloni wish some kind of compensation (e.g., reduction of municipal tax, supply of electricity from biogas). In her opinion, compensation is not the solution. The residents of Tebloni have suffered because of the existing sanitary landfill. This situation poses a threat to their health and safety. Furthermore, although a natural aquifer exists in Tebloni, the water is polluted because of the leachates from the sanitary landfill. So, the residents are supplied by the central water network. A reduction in the price that residents have to pay for water supply could be a possible solution. Overall, stakeholder 1, as the representative of the residents of Tebloni, is very disappointed by the management of solid waste in Corfu and she disagrees with the setting of the new sanitary landfill in Tebloni.

Local political stakeholders were asked if they think that the establishment of a solid waste treatment plant (SWTP) and sanitary landfill will solve the waste management problem of the island. Stakeholders 1, 3, and 4 disagree with the establishment of the SWTP on Corfu. Stakeholders 3 and 4, the NGO representative and the local party activist, stressed that the planned SWTP is not the solution. According to them, the composition of municipal solid waste is 50% organic waste, 35% recyclable materials, and 15% industrial waste. They claimed that three composters are required to manage the organic waste, which reaches 50% of the total waste on Corfu, in connection with proper source separation, to recover 85% of total municipal solid waste.

Stakeholder 2, the waste management expert, suggests that a sanitary landfill is necessary, in addition to the SWTP, because, based on the inverted solid waste management pyramid (prevention, reuse, recycling, energy recovery, landfill of residue), a sanitary landfill is essential for the residual material. So, according to him, the challenge is to divert as few waste residues as possible to the sanitary landfill.

*5.4. The Contribution of Tourism to the Problem of Waste Management*

The problem regarding waste management increases during the high touristic season. Based on the interviews, waste generation on the island of Corfu in winter reaches 150–170 tons per day. During the summer, it comes to 350–400 tons per day because the population increases about tenfold. Corfu can accept about 100,000 visitors according to the capacity of the hotels. As expected, the municipalities are not able to manage these waste volumes. However, the chairman of the hotel owners of Corfu (stakeholder 6) stated that the problem exists during the winter as well. Therefore, tourism contributes greatly, but it is not solely responsible for the waste generation. Moreover, stakeholder 10, a hotel manager, mentioned that infrastructures are still deficient.

At the same time, Stakeholder 1, the representative of the residents of Tebloni, claimed that, "for the last 40 years, tourism stakeholders did not care about the environment and the sustainability of Corfu". She thinks that by introducing the "pay-as-you-throw" principle, tourism firms will have financial incentives to manage their waste properly. Contrary

to her, stakeholder 3, the NGO representative, stated that tourism stakeholders are more sensitized to proper waste management than the citizens.

Stakeholder 5, the Vice Mayor, reported that tourism entities must contribute to the source separation system. Stakeholder 2, the waste management expert, mentioned that tourism entities produce a considerable amount of mixed waste. For a solution to be found, all citizens and stakeholders need to contribute to source separation. There are good practices worldwide that could be implemented, and stakeholder 3, the NGO representative, stated that, "the tourism professionals can contribute to the source separation system as long as there are motives for them".

Stakeholder 9, a hotel manager, stated that there are bins for collecting paper, plastic, and glass all around the hotel that he manages. Furthermore, stakeholder 8, a former hotel manager, mentioned that in the hotel where he worked, they managed food waste and recyclable materials properly due to the requirements of HACCP and ISO 14001. Additionally, stakeholder 6, a chief hotel executive, said that they recycle eight different materials in his hotels and use a composter to manage organic solid waste. Moreover, according to stakeholder 6, his hotels are almost zero waste. Regarding food waste, "foods that are not consumed are given to the staff or are reused to create new dishes". In cooperation with the "Food Bank", a Greek NGO, they are trying to establish an organization in Corfu, which will collect food from hotels and donate it to people in need. Moreover, in one of the hotels that stakeholder 6 manages, the staff has been educated, recycling five materials. Regarding food waste, they create appropriate menus in such a way that they reduce food waste. They carefully manage food supplies so that they do not waste food.

All of the restaurateurs that participated in the present research claim that they take measures for proper waste management. Stakeholder 11, a restaurant owner, said that he separates at source in his restaurants in Athens and Corfu. They do not throw away anything. For example, the bread that has not been used is converted into bread pudding. Fish heads are used to cook fish soup for vulnerable groups. Actually, during the interview, a person belonging to a vulnerable group came and collected food. In addition, stakeholder 15, a restaurant owner, stated that all firms and the local community collect recyclable materials (glass, paper, and aluminum). Regarding food waste, restaurant owner stakeholder 15 believes that an effective solution is the installation of waste composters in each hospitality firm for the proper management of food waste.

## 6. Discussion

A sustainable waste management system consists of three components [36]: (i) the physical components of waste management (e.g., collection and recycling), (ii) the stakeholders involved, and (iii) political, social, and institutional aspects. These aforementioned components have to act in close synergy and harmony for the system to be sustainable. During the summer of 2018, visitors to Corfu were confronted with the enormous amounts of waste that swelled on the island's roads, because the waste management system on the island collapsed. The results of the stakeholders' interviews of our research identified the failure of the physical components of the waste management system, in addition to the political and social confrontation of the local communities located close to the landfills of the island.

In addition, the results of our study indicate that all the stakeholders of Corfu think that the failure of the waste management system to deliver during the summer of 2018 was a major incident that negatively affected the tourist image of the island. This finding is in agreement with what is reported in the literature, i.e., that primary and secondary images influence the perceptions of tourists towards the sustainability of a destination [33]. However, all of them agree that this was only a minor incident for the organic image, as defined by Hankinson [30], of the island, and thus tourism will recover. Indeed, the situation was better in 2020 (compared to 2018) due to the transfer of waste in Kozani, i.e., away from the island, to mainland Greece.

Regarding the physical components of the waste management system, compost projects and "green corners' emerged in Corfu when things came to a dead end. As is apparent from the interviews, all the stakeholders agree that source separation is the solution to the waste management problem in Corfu. However, they are not satisfied with the progress regarding source material separation initiatives. Moreover, they also claim that active citizens' participation in source separation is necessary for any municipality to achieve its goal. This finding is in agreement with what is reported in the literature, i.e., that the contribution of hospitality services to waste generation is impossible to discern from residential or commercial municipal waste indicators [20]. Note that, in Greece, the municipalities are responsible for waste management. A similar situation is reported for the Canary Islands, in Spain [20]. Municipalities finance their waste management activities by charging all households and firms with a municipal tax.

Recycling and source material separation are among the essential processes of the EU waste management hierarchy, a fact identified by all the participating stakeholders. Representatives of the hotels and restaurants stated that they are inclined to contribute to the source separation system. All of the stakeholders representing the tourism sector reported that they care to manage the waste generated from its activities properly. It is also evident from the hotel managers' interviews that organic waste constitutes a significant fraction of the total waste (about 50%). However, on the positive side, the results of our research indicate that all of the hospitality stakeholders have identified the issue of food waste generation and are taking measures towards its prevention and management.

It is challenging for tourism-related organizations to deal with the technical problems of source separation because it requires infrastructure and employee training, as also reported by Diaz-Farina et al. [20]. On top of that, small hotels and restaurants face more difficulties due to the limited available space. All stakeholders also agree that with proper waste prevention and source separation actions, the situation can be improved on the island. Almost twenty years ago, the results of the analysis of Skordilis [23] for the island of Corfu demonstrated that the combination of material recovery at the source with the utilization of the organic fraction is the optimum solution. Moreover, most of the respondents, especially those that are directly involved with tourists, take a proactive approach toward waste management by promoting prevention initiatives.

Regarding the political aspects, all the stakeholders, including representatives of the hotels and restaurants, mentioned that the local authorities do not inform them about the solid waste management in Corfu. Moreover, the professionals involved in tourism are not satisfied and even disappointed because, although they regularly pay the municipal tax, they receive an inferior service regarding waste management by the municipality.

Overall, it is evident that insular tourist hotspots, such as Corfu, both affect the waste management problem and are affected by it. As Eckelman et al. [14] describe, "waste management in the islands is faced with limited land resources, large seasonal fluctuations in waste volumes, and complex social and political dynamics". All these factors have been identified in our results. Waste management should be clearly included among the infrastructure services that a tourist destination has to rely on, to support its organic image. The current model of waste management in Corfu, that relies solely on landfilling, has failed, probably due to its poor design and operation. Therefore, the solution relies on the proper application of the waste management hierarchy, i.e., prevention, source separation and recycling, and landfilling, as the ultimate solution for the residual waste. Towards this goal, the tourism stakeholders, based on our research, seem to have a clear viewpoint. For instance, they have identified the problem that food waste generates and have taken action to prevent it and manage it through composters. It is also evident that hospitality premises that belong to large-scale international firms can be a good driver for turning the whole tourism sector toward more sustainable waste management practices. Finally, our results clearly indicate an expression of the not-in-my-backyard syndrome by the residents of the neighboring areas of landfills. The necessary infrastructure has to be established after thorough consultation with the residents and the tourism stakeholders

on the island. As MacRae and Rodic [36] conclude, "effective waste management systems should take into account local socio-cultural and political-economic factors, which tend to be highly specific".

## 7. Conclusions

In this paper, we conclude that waste management is affected by tourism activities, but it can also severely affect the image of a touristic destination. The situation in Corfu, in Greece, has been presented as a case study since the waste management problem peaks during the high touristic period on the island. A field survey has been conducted to identify the viewpoints of the tourism stakeholders on the waste management problem that the island of Corfu experienced recently. Our results specifically indicate that tourism stakeholders are not satisfied with the municipal waste management services on the island, and they emphasize source separation as the critical factor for proper waste management. Most of them also think that the inefficient waste management on the island is a serious problem that affects Corfu's image as a touristic destination. To preserve the island's organic image, all local stakeholders should engage in sustainable practices that preserve a waste-free environment that the tourists can enjoy.

Regarding the policy implications of our study, local governments and all policymakers should take into account the fact that proper infrastructure, especially for proper solid waste management, is an essential part of sustaining the organic image of a destination. Based on the results of our survey, tourism stakeholders are willing to participate in a collaborative effort with the local municipalities toward sustainable waste management on the island. All the stakeholders in Corfu have to realize that the solution has to be found within the geographical limits of the island; otherwise, it will not be lasting and sustainable.

**Author Contributions:** Conceptualization, A.-S.K. and K.A.; methodology, A.-S.K.; validation, P.-G.T.; investigation, A.-S.K.; data curation, A.-S.K.; writing—original draft preparation, A.-S.K. and K.A.; writing—review and editing, K.A. and P.-G.T.; supervision, P.-G.T. All authors have read and agreed to the published version of the manuscript.

**Funding:** This research received no external funding.

**Institutional Review Board Statement:** Not applicable.

**Informed Consent Statement:** Informed consent was obtained from all subjects involved in the study.

**Data Availability Statement:** Data are available upon reasonable request.

**Acknowledgments:** The authors would like to thank all of the interviewees that participated in the study.

**Conflicts of Interest:** The authors declare no conflict of interest.

## Appendix A

**Table A1.** Full list of stakeholders that participated in the research.

| Stakeholder Number | Stakeholder Description |
| --- | --- |
| 1 | Chairwoman of the cultural association of Tebloni |
| 2 | Waste management expert and scientific associate to the municipality of Corfu |
| 3 | Member of Waste Watch Corfu, an independent NGO |
| 4 | Activist, head of a local political party |
| 5 | Vice Mayor of circular economy and environment of the municipality of North Corfu |
| 6 | Hotel chief executive, chairman of the hotel association of Corfu, member of the Board of Directors of the Greek Tourism Confederation (SETE) |

**Table A1.** *Cont.*

| 7 | Hotel owner |
|---|---|
| 8 | Former hotel manager and marina manager |
| 9 | Hotel manager |
| 10 | Hotel F&B manager |
| 11 | Restaurant owner/chef |
| 12 | Restaurant owner/chef and member of the Board of Directors of the Corfu Gastronomy Association |
| 13 | Restaurant owner and member of the Board of Directors of the Corfu Gastronomy Association |
| 14 | Restaurant owner and member of the Board of Directors of the Corfu Gastronomy Association |
| 15 | Restaurant owner |
| 16 | Tour operator |

**Appendix B**

Questions for hotel and restaurant owners:

1.  Are you satisfied with waste management on Corfu?
2.  Based on your experience, do you think that the problem is increasing throughout the tourist season?
3.  In your opinion, is it possible for a municipality to properly manage waste without source separation?
4.  Are you personally satisfied with the actions that have been taken on Corfu regarding source separation?
5.  Are the citizens sensitive to this issue?
6.  Who do you think must be involved in the process of source separation and, considering the high value of recyclable materials, who should benefit from this value?
7.  In your opinion, should hotels and tourist accommodations/restaurateurs, as a cooperative, organize an autonomous system of source separation and managing the flows of the recyclables or should they just be part of the general system of the municipality?
8.  Do you think that you are sufficiently informed about the issue of the handling of solid waste in Corfu or do you have an empirical vision of the issue?
9.  As a tourism entity, are you inclined to contribute to the source separation system that will be established?
10. Based on your experience, what do you think visitors believe about this issue? Has Corfu′s destination image been damaged because of inefficient waste management?
11. Do you think that poor waste management acts in a deterring manner when it comes to choosing Corfu as a destination?
12. In this firm, do you take measures to properly manage waste, both for material recycling and food waste?
13. Can you deal with the technical problems of source separation within your firm?
14. Do you think that the municipal council tax that you pay for the municipal cleaning service is reasonable?

Questions for experts in waste management:

1.  Based on the negative experience for two decades that we have had at the sanitary landfill of Tebloni, what is your opinion about the placement of a) the SWTP and b) the new sanitary landfill in the area of Tebloni?
2.  What is your opinion about the restoration of the sanitary landfill of Tebloni that was caused by the unacceptable handling of the previous years? Do you agree with the existing restoration study which foresees a deposit of 40,000 tonnes more than what has been already transferred there?
3.  Due to the very serious burden throughout the previous period and the possible inconveniences from activities planned there, what do you think about the remunerative benefits to the residents of Tebloni and what can they be?

4. Was there a specific action plan regarding the source separation since the merger of the municipalities?
5. The municipality of North Corfu is so far the only municipality of our island that has proceeded to certain specific actions regarding the source separation. Which were those actions? Are you planning more steps in the direction of source separation?
6. In your opinion, is it possible for a municipality to properly manage waste without source separation?
7. Do you think that the general direction of solving the problem of solid waste management in our country, and in Corfu (solid waste treatment plant and sanitary landfill), is compatible to increase the percentage of solid waste which is recycled?
8. Are you personally satisfied with the steps that have been done in Corfu regarding source separation?
9. Who do you think must be involved in the process of source separation and, considering the high value of recyclable materials, who should benefit from this value?
10. In your experience, do you think that the problem is intensified throughout the tourist season?
11. If yes, do you think that the stakeholders are inclined to contribute to finding a solution?
12. Do you think that tourism professionals (accommodation, catering, entertainment) can make a significant contribution to source separation and if so, what could their role be?
13. What steps does Corfu have to take to increase source separation more rapidly?
14. In your opinion, is there proper information about the benefits of recycling to stakeholders as well as to citizens?

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
