# Peer review of "Understanding the Impact of Waste Management on a Destination′s Image: A Stakeholders′ Perspective"

_tourismhosp, doi:10.3390/tourhosp4010004_

Round 1

Reviewer 1 Report

The article deals with the topic of waste management in the world-famous tourist destination - Corfu. The article is a qualitative study based on interviews with 16 key actors. Although this sample is not very large, it is sufficient for the study.

The research presented in the article is well-designed, the methodology is clearly explained, and the results are adequately presented. However, the article has some shortcomings, the elimination of which would certainly increase the attractiveness of the article among readers.

First, the article lacks any graphic information. At the same time, the article describes various problems that have affected waste management in Corfu in recent years. Photos from this period would surely enliven the text of the article. Similarly, the article describes the activities of NGOs that responded to this situation. In this case, it would be appropriate to include posters, print screens from social networks, etc., so that it is clear what these NGOs were striving for and how they presented their intentions.

Second, the reader learns practically nothing in the Conclusions chapter. It is clear that waste management is essential. But what specific conclusions did the article reach? I would expect this chapter to summarize the results presented in Chapter 5. I would recommend the authors rewrite this chapter.

Author Response

The article deals with the topic of waste management in the world-famous tourist destination - Corfu. The article is a qualitative study based on interviews with 16 key actors. Although this sample is not very large, it is sufficient for the study.

The research presented in the article is well-designed, the methodology is clearly explained, and the results are adequately presented. However, the article has some shortcomings, the elimination of which would certainly increase the attractiveness of the article among readers.

We thank the reviewer for the encouraging comments.

First, the article lacks any graphic information. At the same time, the article describes various problems that have affected waste management in Corfu in recent years. Photos from this period would surely enliven the text of the article. Similarly, the article describes the activities of NGOs that responded to this situation. In this case, it would be appropriate to include posters, print screens from social networks, etc., so that it is clear what these NGOs were striving for and how they presented their intentions.

A picture has been added within the text. However, readers can check the images within the websites from the various media reported in the references of the manuscript.

Second, the reader learns practically nothing in the Conclusions chapter. It is clear that waste management is essential. But what specific conclusions did the article reach? I would expect this chapter to summarize the results presented in Chapter 5. I would recommend the authors rewrite this chapter.

Both the Discussion and Conclusions sections have been rewritten to better summarize the results.

Reviewer 2 Report

The paper focuses on a very interesting theme. The goal of the paper is clear and well-motivated. The authors investigate the situation in Corfu, in Greece, as a case study. A field survey has been conducted to identify the viewpoints of the tourism stakeholders on the waste management problem.

The review of literature regarding the impact of waste management on destination image is insufficient, and need to be improved to strengthen the logical background of the paper.  

The paper currently does not clearly show what this research contributes beyond existing research. The current discussion on contributions needs to be expanded and enhanced. The discussion needs more detailed description in line with previous research.

I wish the authors the best of luck.

Author Response

The paper focuses on a very interesting theme. The goal of the paper is clear and well-motivated. The authors investigate the situation in Corfu, in Greece, as a case study. A field survey has been conducted to identify the viewpoints of the tourism stakeholders on the waste management problem.

We thank the reviewer for the encouraging comments.

The review of literature regarding the impact of waste management on destination image is insufficient, and need to be improved to strengthen the logical background of the paper.

The literature review has been enhanced, and more relevant references have been added:

Willmott, L.; Grasi, S. Solid Waste Management in Small Island Destinations: A Case Study of Gili Trawangan, Indonesia, Téoros [Online], 31, 3 (HS) | 2012, Available online: http://journals.openedition.org/teoros/1974 (accessed on 10 January 2023). 

Almeida-Santana, A.; Moreno-Gil, S. Perceived Sustainable Destination Image: Implications for Marketing Strategies in Europe. Sustainability 2019, 11, 6466. https://doi.org/10.3390/su11226466 

The paper currently does not clearly show what this research contributes beyond existing research. The current discussion on contributions needs to be expanded and enhanced. The discussion needs more detailed description in line with previous research.

Both the Discussion and Conclusions sections have been rewritten to better summarize the results.